# Exploring the Trade-Off between Model Complexity and Numerical Precision for Efficient Edge AI Inference

## Abstract

When considering the compression of neural networks, the adoption of low-bit representations for both parameters and activations has demonstrated significant efficacy. The process of learning quantized weights through Quantization Aware Training (QAT) stands out as a powerful means to substantially diminish the memory requirements for a specific model to efficiently perform inference. However, despite the numerous works reporting the gains achieved using QAT, a comparison with a notably simpler technique - reducing the model's complexity using fewer parameters - is often absent.

In this paper, we attemp to answer a seemingly simple question: to reduce a given model's storage requirements, is it better to reduce the number of parameters in the model or to reduce the numerical precision? We explore the trade-off between the dimensionality of parameters and activations one can afford to keep in memory, and the numerical precision used to represent them. Through our experiments in image classification, keyword spotting and language modelling, our results suggest that quantizing weights to 2 bits and keeping a high number of parameters seems optimal, regardless of the task considered and model architecture.

## 1 Introduction

Compressing neural networks is often necessary when seeking on-device implementation of artificial intelligence application (Han et al., 2016). When it comes to compression, using low-precision parameters and activations *via* quantization has proven to be an effective way to reduce the memory a model needs to perform inference, while maintaining a good performance at the task it solves (Jacob et al., 2018). Another possibility to fit a very large model into a memory-constrained device is to reduce the number of parameters by scaling the model down. Scaling a model can be done by depth, width or input resolution, or some combination of these factors. This idea was at the core of EfficientNet (Tan & Le, 2019), and it is now commonly accepted that scaling the resolution, depth and width of a model simultaneously yields better results than acting on either of these levers alone.

We propose to combine these two approaches, and ask ourselves the question: given a memory footprint constraint, is it preferable to scale the dimensions of a model and keep a high numerical precision (usually, 32 bits), or is it better to keep a high number of parameters and quantize them to a low numerical precision (8 bits or less)? In this work, we aim at answering this question by considering several tasks and model architectures, and observing how, given a fixed memory footprint, the model accuracy varies when the numerical precision varies jointly with the number of parameters. Our work develops ideas from EfficientNet (Tan & Le, 2019) to explore the trade-off between model complexity and numerical precision. For instance, halving the numerical precision makes it possible to store twice as many parameters at a constant memory footprint.

Current compression methods commonly use 8-bit representations (Jacob et al., 2018; Yang et al., 2020) for several reasons: it incurs no drop in accuracy, 8-bit integer arithmetic is supported on common hardware platforms, and it consumes much less energy. Our work aims at questioning whether 8-bit is optimal or if other options could yield better results. Our results suggest that 2-bit quantized networks offer the best compromise between numerical precision and model complexity, advocating for the development of dedicated hardware (Qiu et al., 2016; Conti et al., 2023).

In section 3, we shall detail how we propose to examine the trade-off between numerical precision and model complexity. In section 4, we will present experimental results on four datasets: image classification on CIFAR-10 (Krizhevsky et al., 2009) and ImageNet (Russakovsky et al., 2015), language modeling on WikiText-103 (Merity et al., 2017), and keyword spotting in the Speech Commands dataset (Warden, 2018).

## 2 RELATED WORK

### MODEL SCALING

Scaling a model means increasing or decreasing its number of parameters, and in a broader sense its computational cost. It can typically be done by adding more layers (depth scaling) as it is commonly done in ResNet (He et al., 2016), by considering wider layers (width scaling) as in WideResNet (Zagoruyko & Komodakis, 2016), or by taking larger signal as input (resolution scaling), typically by considering larger images (see Tan & Le (2019), section 3.2). This opens the question of an optimal compromise between these three directions when scaling a convolutional network. EfficientNet (Tan & Le, 2019) studies this trade-off by appling it to a MobileNet (Howard et al., 2017). The key finding of this paper is that scaling a model by depth, width and resolution simultaneously provides better results than scaling by one of these factors alone.

The same type of research was applied to transformers, either vision transformers or language models. Most works study how scaling a model up increases its performance at certain tasks (Rae et al., 2021; Chowdhery et al., 2023), notably at few-shot learning, but studying how scaling down impacts performance has yet to be done. Here, the available levers for scaling a model are the embedding dimension (which is often equal to the key/value dimension), the number of attention heads and the number of layers.

### NEURAL NETWORK QUANTIZATION

Quantization is the process of constraining the model parameters (and possibly activations also) to a discrete, finite set. Typically, quantizing a tensor (parameter or activation) to $b$ bits means constraining all of its values to lie in a set of $2^b$ elements. Now, quantizing a model can be done using three different approaches.

Quantization-Aware Training (QAT) quantizes parameters iteratively during training. It requires storing the parameters in high numerical precision (32 bits) and quantizing them at each forward pass, accumulating gradients in the high-precision weights. These approaches have demonstrated an ability to quantize weights to low numerical precisions (below 4 bits) without significant drops in accuracy (Guo et al., 2022; Choukroun et al., 2019; Esser et al., 2020; Sun et al., 2020). They tend to be the approaches that perform the best at inference time.

Post-Training Quantization (PTQ) is the process of quantizing a trained model, without re-training the quantized weights - or with a slight finetuning. These approaches are often done by default in edge AI platforms, with a numerical precision commonly reduced to 16 bits (Demidovskij & Smirnov, 2020), 8 bits (Kluska & Zieba, 2020) or even 4 bits (Banner et al., 2019). Yet, further lowering the numerical precision via PTQ yields a degradation in accuracy, and this approach tends to produce poorer results than QAT to obtain models with low-precision parameters.

Finally, some works perform fully quantized training, using sub-32 bits precision only, which involves a floating-point format at 16-bit (Narang et al., 2018) or 8 bits (Wang et al., 2018) precision, or using directly 8-bit integer formats (Wang et al., 2022). Such approaches have the advantage of reducing the memory and computational cost of training a model, and open up the possibility to perform training on chip.

### HARDWARE-AWARE NEURAL ARCHITECTURE SEARCH

Neural Architecture Search (NAS) (Elsken et al., 2019; White et al., 2023) is a field of deep learning which develops methods to automatically find *good* designs of neural architectures for a given task. These designs are searched in a restricted (but possibly infinite) search space which consists of a broad set of architectures. Different architectures from the same search space typically share the

same elementary computational block (for instance, a convolutional layer, a residual block (He et al., 2016), or an inverted bottleneck MBConv (Howard et al., 2017)) which is repeated or dilated with different sizes in each architecture. The goal of NAS is then to find a *good* (or the best) architecture within the defined search space. Early NAS approaches involved large search spaces explored with reinforcement learning or evolutionary algorithms, as per Zoph & Le (2017). The high cost of NAS was later reduced with the introduction of differentiable NAS (Liu et al., 2019) and its continuous optimization approach.

A further direction for NAS integrates hardware constraints, resulting in the subfield of Hardware-aware NAS (HaNAS) (Benmeziane et al., 2021; Zhang et al., 2020). The core of the EfficientNet work (Tan & Le, 2019) can be considered a part of it, as it optimizes the dimensionality of a predefined type of block in a model, where the model should satisfy hypothetical hardware constraints.

## 3 PROPOSED METHOD

### MODEL SCALING

Given its topology, the number of paramaters of a model can be known *a priori*. Also, knowing the dimension of the data it will take as an input, the size of all intermediary calculations (or *activations*) can be known in advance. Since inference does not require storing all activations in memory but only computing them sequentially, knowing the size of the largest activation gives a good estimate of the memory needed during inference. Adding the memory required from the model's parameters and from the largest activation, it is thus possible to estimate the memory a model will require to perform inference, hereafter referred to as its *memory footprint*.

Now, there are several ways to vary the memory footprint of a model. A breakthrough approach when it was released, EfficientNet (Tan & Le, 2019) details three levers for computer vision:

- The *depth* of the model, denoted $d$, that is, the number of layers it comprises;
- The *width* of the model, denoted $w$, typically the number of output neurons in a linear layer or the number of filters in a convolutional layer;
- The *resolution* of the input data (specifically, an image), denoted $r$. It is seen as a multiplicative factor on all dimensions of the input signal, and will impact the size of all activations. For instance, it will change the dimensions of an input image (by a factor $r$) or the sampling frequency of an audio signal, but it cannot be transposed to natural language processing.

Varying the number of parameters using any of these levers is called *scaling*. An important insight from EfficientNet is that scaling is optimal when performed on these three levers at the same time, which they call compound scaling. We propose a visualization of different scaling methods in Figure 1. They also remark that the number of parameters in the model is proportional to $d$, $w^2$ and $r^2$. In our study, since some of the tasks we consider do not involve images, we will not consider scaling the input resolution $r$.

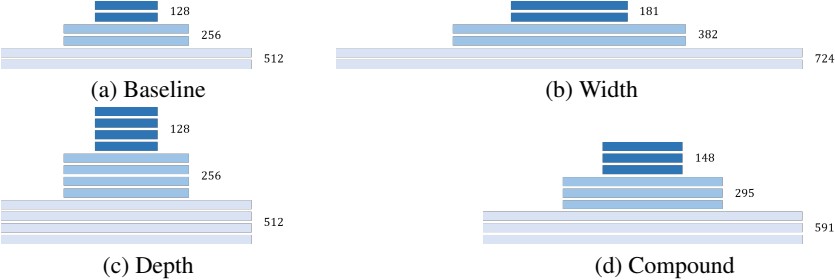

Figure 1: Scheme of different scaling methods applied to a fictional convolutional network. Assuming the baseline model has N parameters, each of the scaled models has 2N parameters. On the side is an example of output convolutional filters number in each block.

When dealing with transformers, as illustrated in Figure 2, one can perform analogous scaling by leveraging the following settings:

- The embedding dimension, that is the dimension of the space used to represent tokens;

- The number of attention heads;

- The number of layers in the decoder.

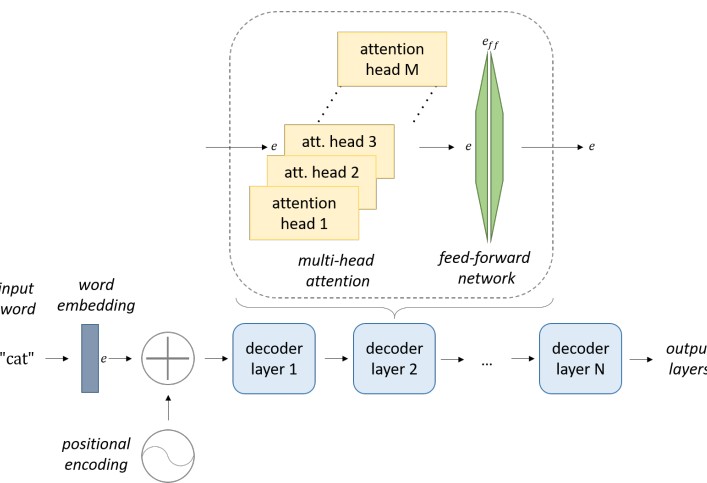

Figure 2: Simplified scheme of a decoder-only transformer architecture exhibiting the levers on which to play for scaling: the embedding dimension $e$, the number of attention heads $M$ and the number of decoder layers $N$.

### ADDING NUMERICAL PRECISION TO THE SCALING SCHEME

In this work, we aim at extending the ideas of EfficientNet-like compound scaling by integrating the numerical precision into the scope of the analysis, and to apply it to tasks other than image processing. Considering a fixed type of neural network, a generic task where one wishes to maximize some $Performance$, and denoting $\mathcal{N}(w, d, b_p, b_a)$ the network with width $w$, depth $d$, numerical precision of parameters and activations $b_p$ and $b_a$, our target can be formulated as the following optimization problem:

$$\max_{d,w,b_p,b_a} \quad Performance\left[\mathcal{N}(w, d, b_p, b_a)\right]$$
$$\text{s.t.} \quad \text{Memory footprint}(\mathcal{N}) \leq \text{target memory} \tag{1}$$

Due to the variety of possible combinations, we followed the approach from EfficientNet (Tan & Le, 2019). We first built a baseline with high-precision, 32-bit parameters and activations that respected the task-related memory constraint. This reference model with width $w_0$ and depth $d_0$ was then scaled using the compound scaling, multiplying its width and depth by $\alpha^{1/3}$ when seeking to scale the model number of parameters by $\alpha$. To simplify the problem, we set the activations' precision to a fixed number of bits above the precision of parameters, that is:

$$b_a = b_p + k$$

for some value of $k \in \{0, 1, 2\}$. Now, lowering the numerical precision from 32 to $b_p$ bits typically allows to store $32/b_p$ times more parameters; consequently, our scaling factor will be set to $\alpha \approx 32/b_p$. Our problem of interest now consists in jointly varying the number of parameters and the numerical precision of weights, and can consequently be reformulated as follows:

$$\max_{b_p} \quad Performance(\mathcal{N}_\alpha)$$

$$\text{with} \quad \alpha := \underset{a \in \mathbb{R}_+^*}{\text{argmax}} \text{ Memory footprint}(\mathcal{N}_a, b_p)$$

$$\text{s.t.} \quad \text{Memory footprint}(\mathcal{N}_a, b_p) \leq \text{target memory} \tag{2}$$

$$w_\alpha := \alpha^{1/3} w_0, \ d_\alpha := \alpha^{1/3} d_0$$

$$\mathcal{N}_\alpha := \mathcal{N}(w_\alpha, d_\alpha, b_p, b_p + k)$$

In plain language, it means that for every possible parameter bitwidth $b_p$, we shall consider the largest possible scale $\alpha$ satisfying the memory constraint and evaluate the performance of the resulting model. The bitwidth having the best performance will then be identified as the best suited for the considered task.

MODEL QUANTIZATION VIA LSQ

To obtain models having low-bit parameters and activations, we used quantization-aware training (QAT) because it delivers consistently good results. Among many possibilities when it comes to QAT methods, we used Learned Step size Quantization (LSQ, Esser et al. (2020)) because it is simple to implement, relies on simple operations (rounding and multiplications) and does not depend on exogenous hyperparameters one might have to finetune. The method proposes to quantize any scalar $x \in \mathbb{R}$ to $b$ bits depending on a scaling factor $s \in \mathbb{R}_+^*$ by mapping it to values in a segment $S = \{Q_N, ..., Q_P\}$. If $x$ can be assumed to be positive (i.e. ReLU activation), these can be set as $Q_N = 0$ and $Q_P = 2^b - 1$. If $x$ is a signed tensor then we shall define $Q_N = -2^{b-1}$ and $Q_P = 2^{b-1} - 1$. In both cases, the discrete segment $S$ contains $2^b$ values and can be encoded using $b$ bits. The quantization counterpart of $x$ is then defined as

$$q_s(x) = \begin{cases} sQ_N & \text{if } x/s < Q_n \\ s\lfloor \frac{x}{s} \rceil & \text{if } Q_N \leq x/s \leq Q_P \\ sQ_P & \text{if } x/s > Q_P \end{cases} \tag{3}$$

$$= s \, \text{clip}\,(x/s, Q_N, Q_P)$$

The backward rule of LSQ follows the spirit of the Straight-Through estimator (STE, Bengio et al. (2013)) with

$$\frac{\partial}{\partial x} q_s(x) = \begin{cases} 1 & \text{if } Q_N \leq x/s \leq Q_P \\ 0 & \text{else} \end{cases} \tag{4}$$

and defines the derivative of $q_s(x)$ with respect to $s$ as

$$\frac{\partial}{\partial s} q_s(x) = \begin{cases} -\frac{x}{s} + \lfloor \frac{x}{s} \rceil & \text{if } Q_N \leq x/s \leq Q_P \\ Q_N & \text{if } x/s < Q_N \\ Q_P & \text{if } x/s > Q_P \end{cases} \tag{5}$$

## 4 EXPERIMENTAL RESULTS

### 4.1 CIFAR-10

CIFAR-10 (Krizhevsky et al., 2009) is a lightweight image classification task of $32 \times 32$ pixel images depicting 10 different object classes. It comprises 50k training images and 10k test images. We tried two different models on this dataset: a simple ResNet-20 as described in (He et al., 2016), and an EfficientNet-type network (Tan & Le, 2019) which we designed with the same number of parameters as the default ResNet-20, that is about 270k parameters, which is much less than in the original EfficientNet paper (see appendix A.1.2 for more details). In order to make quantization

easier, and to avoid assigning a bit for the sign of activations, we chose to use activations taking positive values. Hence, we set the activation functions of the EfficientNet to ReLU instead of SiLU. Both models were trained at different compression ratios: their memory footprint was decreased by a factor $2, 5$ and then $10$ in the experiments. By construction, models specified at a given compression ratio all have a very similar memory footprint. Since the images size is $32 \times 32$ pixels, we chose not to apply any resolutions scaling ; hence, the compound scaling for this part applies only to width and depth.

In all cases, we performed a 10-fold cross validation over 200 epochs. A first round of experiments was conducted with ResNet-20 only and a compression in width (see Figure 3a). Training was done as in the original ResNet paper (He et al., 2016), that is with a SGD optimizer, learning rate of $10^{-1}$ and batch size of $128$, momentum $0.9$ and weight decay $10^{-4}$, and a learning rate divided by 10 at iterations $32k$ and $48k$.

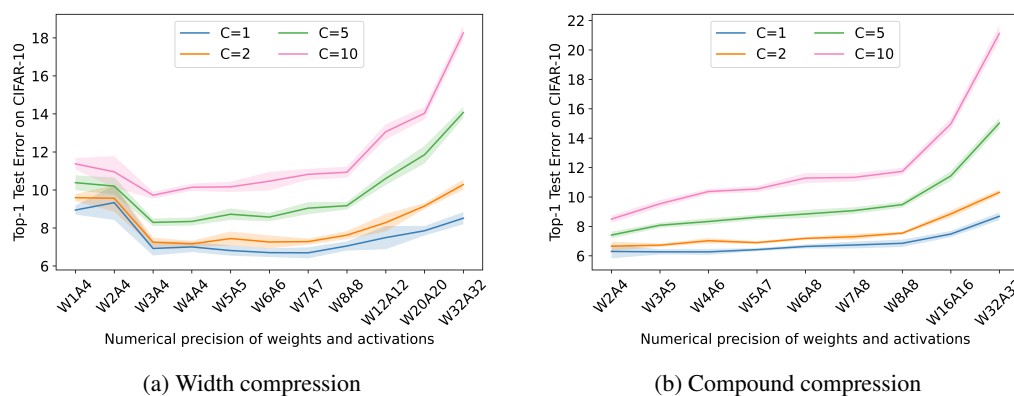

(a) Width compression  (b) Compound compression

Figure 3: Test error rates on CIFAR-10 with a ResNet-20-based model at different compression rates, using width compression (left) or compound compression (right), depending on the numerical precision of weights and activations. Areas in light colors represent the standard deviation of test errors. WXAY denotes a model with parameters encoded in X bits and activations in Y bits. "C=r" denotes a model compressed "r" times. Hence, the point at abscess "W4A6" on the "C=5" line reports the accuracy of a model with weights and activations quantized to 4 and 6 bits respectively, such that its memory footprint is 5 times lower than the baseline model.

As seen in Figure 3a, when performing scaling with respect to the model width alone, this experiment suggests that the optimal trade-off between model size and numerical precision is with weights quantized at 3 bits (and 10 times as many parameters as the 32-bit baseline) for most compression ratios. The only exception is when considering an uncompressed network (C=1) where higher numerical precision (here, weights quantized to 6 bits) yields higher classification accuracy.

In the second round, we applied a compound compression to reduce the memory footprint (as per equation 2), and implemented both ResNet-20 and our EfficientNet. This time, we used Adam as the optimizer with a $10^{-3}$ learning rate, an effective batch size of 1024 (128 over 8 GPUs) and a learning rate division by 2 on plateau.

Now using compound scaling, as shown in Figure 3b, the optimal trade-off seems to be with the lowest numerical precision possible (and as many parameters as possible) for all compression ratios considered. Note that the results for 1-bit are missing, due to the fact that the number of high-precision latent parameters is so much larger when quantizing to 1 bit that the training time becomes prohibitive for low compression ratios. Thus, the run time in these cases exceeded the allowed maximum we set (that is, one week).

## 4.2 IMAGENET

The well-known ImageNet dataset (Russakovsky et al., 2015), also called ISLVRC-12, is a heavier image classification task. It comprises about 1.3 million train images and 45k validation images, with 1000 image classes. Due to the difficulty of evaluating predictions on the actual test images on

the ImageNet server, we followed the similar split as (Tan & Le, 2019), that is we considered the provided validation set as the test set and randomly selected 25k images from the training data as the validation dataset used to determine the best epoch. The reported test error is thus the error on the original "validation" set, which the model never saw even during our validation steps.

The default transform applied in this dataset are a random resize of the smaller dimension of the image between 256 and 480 pixels and then a random crop of $224 \times 224$ pixels is applied, yielding the image used for training.

The model we applied here is an EfficientNet (Tan & Le, 2019) whose memory footprint is 10 times as low as the original EfficientNet-B0. The basis model having memory requirements of 26 MB to infer on a single $224 \times 224$ image, we considered a maximum memory budget of 2.6 MB. An important point is also that, for the sake of simplicity and comparison with other experiments, we chose not to scale the model by input resolution, contrary to the original EfficientNet approach. Additionally, we changed all activation functions in the EfficientNet to ReLU instead of the original SiLU. In fact, activations resulting from SiLU have many, low absolute-value negative entries, thus leading to unnecessarily using half of the allowed bits to encode these values while yielding higher error in positive values. On the contrary, ReLU outputs positive tensors with many zero-valued entries, which can perfectly be mapped in a discrete interval $\{0, ..., 2^{b-1}\}$ which also has a zero. Yet, similarly to the original EfficientNet approach, all reported experiments were obtained using compound scaling on the model's width and depth: the 32-bit, $10\times$ compressed model baseline was obtained by multiplying EfficientNet-B0 depth and width by a factor $10^{1/3}$.

Our model was trained over 30 epochs on ImageNet using an effective batch size of 1024 (128 over 8 GPUs), using the Adam optimizer, a learning rate of $10^{-3}$. Despite the parameters having different sizes, we loaded as many of the EfficientNet-B0 pre-trained weights as could fit in the model, with the intuition that the skip connections present in the architecture could help perform better than starting from scratch.

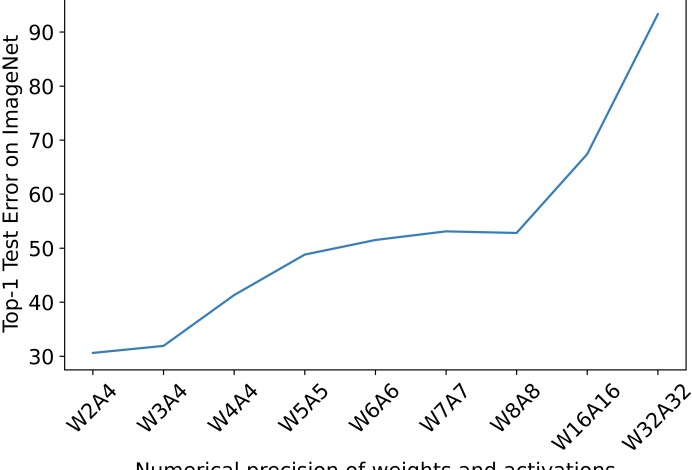

Figure 4: Top-1 test error on ImageNet (ILSVRC 2012) with an EfficientNet-B0 compressed at a ratio of 10 using compound scaling, depending on the numerical precision of weights and activations.

Again, this experiment (see Figure 4) suggests that the optimal trade-off between the model complexity and numerical precision of its parameters occurs with the lowest numerical precision (and the greatest number of parameters) possible.

### 4.3 WIKITEXT-103

WikiText-103 (Merity et al., 2017) is a corpus of Wikipedia articles, comprising over 100 million tokens for a vocabulary of size 270k. This dataset serves as a benchmark for language modelling tasks, where the goal for the model is to predict the next word in the text given a sequence of words.

We chose to apply a NLP Transformer model to this task because of the state-of-the-art results of this type of model. More specifically, we trained a Transformer model with adaptive inputs (Baevski & Auli, 2019). This choice is motivated by the relatively limited number of parameters in the model (270 million) compared to more recent state-of-the-art models having several (and up to hundreds of) billions of parameters: see for instance RETRO (Borgeaud et al., 2022) which has 7.5 billion.

For the scaling of such a model, we followed the spirit of EfficientNet and varied jointly the embedding dimension, the number of attention layers and the number of final layers. In the perspective of edge implementation, we started from a $32\times$ compression ratio, such that the 1-bit quantized model will have the same number of parameters as the baseline one. This means that the compressed full-precision transformer we consider should not have more than $8.4$ million parameters, which is remarkably small for a transformer.

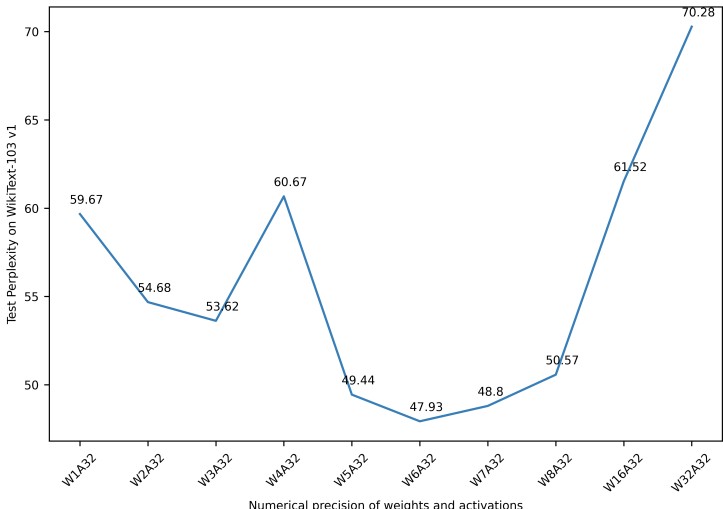

Figure 5: Test perplexity on WikiText-103 v1 using a Transformer with adaptive inputs with weights quantized at different precisions.

As showed in figure 5, the conclusion is not quite as clear as when working with the previous datasets. It seems that the more irregular distribution of weights in transformers (Maekaku et al., 2022) than in convolutional networks resulted in a significant degradation in performance. Also, in this experiment, the activations were not quantized, as we found that quantizing them produced a very large degradation in performance (see appendix A.3.1 for more details). In fact, the distribution of activations is so irregular in transformers that some approaches such as Xi et al. (2023) suggest to change the representation basis of activations and quantize them in this different basis. We also found that LSQ was ill-suited for weight quantization above 4 bits, and we thus applied plain linear quantization above this point. Thus, we can draw an insight from this experiment which goes in the same direction as the previous ones: up to a certain point, lowering the numerical precision of weights and keeping a relatively high number of parameters seems best.

## 4.4 KEYWORD SPOTTING

The Google Speech Commands dataset (Warden, 2018) is made of audio recordings of 34 different keywords pronounced by different speakers. These keywords are simple speech commands such as "yes", "no", "left", etc. sampled at 16 kHz. The training dataset is made of 84k audio samples, the validation set 10k, and the test set 11k.

A common task on this dataset is to predict the label of the speech command. To this aim, we first transformed each speech command to a 2d image by applying a mel-frequency cepstrum (MFC) transform on which we retained the 40 most significant coefficients, yielding a $40 \times 81$ image representation of the keyword. This 2d image representing the command's frequency was then fed to a ResNet. By default, we considered a ResNet with a low memory budget, that is 230 kB.

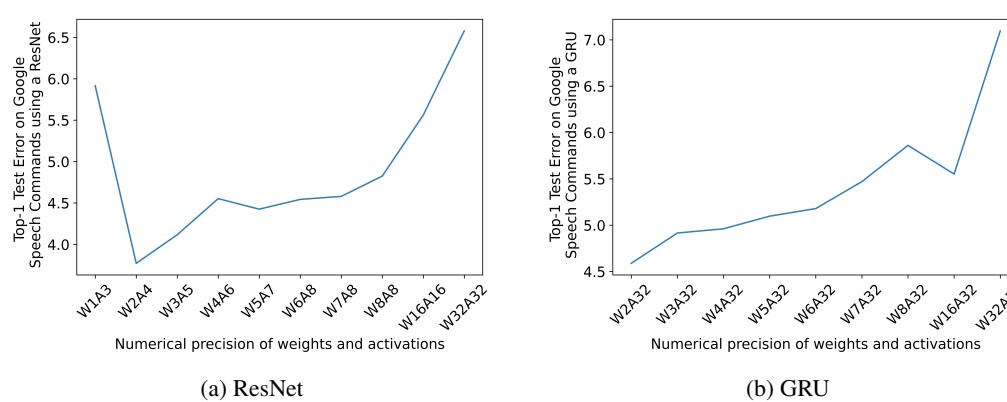

(a) ResNet              (b) GRU

Figure 6: Top-1 test classification error on Google Speech Commands with a ResNet (left) and GRU (right), with weights quantized at different precisions

Here again (see Figure 6a), a lower numerical precision combined with a high number of parameters yields the best results, with 2 bit being the optimal trade-off. Interestingly, 1-bit quantization does not yield the best performance in this case. To validate our approach, we trained a GRU network with the same memory budget as above (230 kB). Here again, and for the sake of simplicity, we performed weights-only quantization. To scale the model, we jointly increased the number of layers and the hidden dimension. Note that we removed the 1-bit quantization as the error rate was much higher.

Once again (see Figure 6b), chosing a model with the lowest numerical precision and highest number of parameters delivers the best results, with 2 bits quantization appearing as the best trade-off.

## 5 CONCLUSION AND DISCUSSION

Without any ambiguity, our work suggests that models having a high number of parameters in low numerical precision perform better than those with fewer parameters in higher numerical precision, at least to some extent. For all experiments involving a convolutional neural network or a GRU, we found that compressing the model via 2-bit quantization and compound scaling is preferable to any other choice in the cases we studied. This was particularly the case when considering compressed models, and even more so when the compression ratio was high. Yet, we must bring a slight nuance to this claim, as our experiments on CIFAR-10 (see Figure 3a) suggested that 2-bit quantization was not always optimal when using width-only scaling. It also appears that compound scaling generally gives better results than width-only scaling, in line with the insights from EfficientNet (Tan & Le, 2019). However, at full precision, compressing the ResNet-20 model $5\times$ or $10\times$ delivered worse results when using compound scaling instead of width-only scaling. This observation calls for more advanced investigation of methods to scale models *down*, not only *up* as most existing methods propose. Yet, our experiments have some limitations:

- **Scaling methods** We tried to replicate the scaling methods presented in EfficientNet. Yet, this method was not designed to scale models *down* but rather *up*. Thus, it is possible that other scaling methods could yield a better performance, particularly for high compression ratios at high numerical precision (that is, few parameters). Investigating scaling methods for model compression could help better understand how reducing the number of parameters in a given model type impacts its performance.

- **LSQ for 6- to 8-bit quantization** As our experiments suggest, LSQ might not be a great quantization method for 6- to 8-bit quantization. Indeed, we suspect that multiplying incoming gradients by the highest integer possible, as per equation 5, yields unnecessarily large gradients. In the future, we plan to extend the experiments to other QAT methods.

- **Focus on memory vs. FLOPS** Contrary to EfficientNet, our work focuses exclusively on the memory space taken by a model when performing an inference. It does not consider the

number of operations it requires at all, which could give better insights to design dedicated hardware.

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

## A    ADDITIONAL EXPERIMENTAL DETAILS

### A.1    BASELINE MODELS

#### A.1.1    PERFORMANCE OF BASELINE MODELS

This subsection aims at simply presenting the memory footprint of all models used in our experimental results in section 4.

Table 1: Performances of all baseline models used in our experiments. All models were scaled during our experiments, by factors varying from 2 to 30, which explains the difference in performance between the baseline model and our reported results.

| Dataset | Model | # param. | Metric | Score |
|---|---|---|---|---|
| CIFAR-10 | ResNet-20 | 270k | Top-1 test error (%) | 8.5 |
| CIFAR-10 | EfficientNet (light) | 270k | Top-1 test error (%) | 6.5 |
| ImageNet | EfficientNet-B0 | 5.3M | Top-1 test error (%) | 22.3 |
| Google Speech-Commands | ResNet-20 | 270k | Top-1 test error (%) | 4.6 |
| Google Speech-Commands | GRU | 280k | Top-1 test error (%) | 5.3 |
| WikiText-103 | Transformer (adaptive inputs) | 247M | Test perplexity | 18.7 |

#### A.1.2    DIMENSIONALITY OF BASELINE MODELS

Table 2: Number of parameters and dimensionality of the largest activation during inference for different models and scaling ratios.

| Model | Scale ratio | Input size | # param. / Act. dim. |
|---|---|---|---|
| ResNet-20 | 1 (baseline) | 32 | 270k / 16k |
| EfficientNet (light) | 1 (baseline) | 32 | 254k / 49k |
| EfficientNet-B0 | 1 (baseline) | 224 | 5.3M / 1.2M |
| ResNet-18 | 1 (baseline) | 224 | 11.7M / 0.8M |

### A.2    IMAGENET IMPLEMENTATION

Standard ImageNet data augmentations were used during our training (He et al., 2015). More precisely, during training, images were randomly resized between $240$ and $480$ pixels (on their smaller dimension), and then a random crop of $224 \times 224$ pixels was extracted to provide the actual training image. During testing, images were also randomly resized randomly between $240$ and $480$ pixels, then 5 crops (center and the four corners of the image) were extracted from the image, together with the 5 crops from the horizontally flipped image, yielding 10 crops of the test image. Then, predictions were averaged on the 10 crops.

## A.3 ADDITIONAL RESULTS

### A.3.1 TRANSFORMER WITH ADAPTIVE INPUT REPRESENTATION ON WIKITEXT-103 QUANTIZING WEIGHTS AND ACTICATIONS

In our experiments, we also tried to quantize the activations (together with the weights) of the transform model with adaptive inputs. The results, as reported in figure 7, show very poor performance when the numerical precision diminishes significantly. Also, it exhibits a rather erratic behavior from which we can hardly draw conclusions. We suspect this difficulty when quantizing activations could come from the very irregular distribution of activations in a transformer, which is far less smooth than in a convolutional model; thus, significant clamping of values due to quantization range may incur large losses of information.

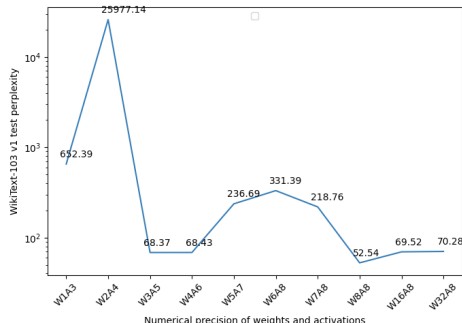

Figure 7: Test perplexity on WikiText-103 v1 using a Transformer with adaptive inputs with weights and activations quantized at different precisions.

