# OpenReview forum: "Exploring the Trade-Off between Model Complexity and Numerical Precision for Efficient Edge AI Inference"
_ICLR.cc/2025/Conference — Submitted to ICLR 2025_

### Official Review · Reviewer_9xBC · 2024-10-31

**Soundness:** 3
**Presentation:** 2
**Contribution:** 1
**Rating:** 3
**Confidence:** 4

**Summary:**

This paper discussed the scaling law of several factors including network widths, depths, and a new factor precision. The authors primarily conduct experiments on relatively small-scale datasets and tasks. They found that with 2-bit weight quantization, they can achieve largest accuracy improvement.

**Strengths:**

1. Number precision is a valuable attribute when scaling larger networks. For example, today's LLMs always use FP16 form and sometimes INT8/FP8 to reduce memory.

2. The authors verified their setup on different types of datasets.

**Weaknesses:**

1. The experiment setup is not very fair and practical. Some of the precision setups are not deployable on modern GPUs, especially W3A3, W5A5, and W6A6. Network width, depth, and input size are not a problem on GPUs. Therefore, this precision choice limits its size.

2. The LSQ training algorithm, or broadly, QAT algorithms, requires FP32 weights memory, gradient memory, and optimizer memory. They will significantly increase the training costs if you lower your precision (so that you can have more parameters under the same budget). This is not considered in the paper. Also, given the performance of QAT, I am not surprised to see the results, as maintaining FP32 weights in training can significant help the convergence and achieve near floating-point accuracy.

3. The experiment scale is this paper is too small compared to real scaling-law study like EfficientNet, LLMs.

3. The structure and the visualization of this paper are somehow lower than the average quality of the conference papers. I encourage some improvement in the presentation of the paper.

**Questions:**

N/A

---

### Official Review · Reviewer_25rx · 2024-11-01

**Soundness:** 2
**Presentation:** 3
**Contribution:** 3
**Rating:** 6
**Confidence:** 4

**Summary:**

The paper studies the trade-off between reducing the number of model parameters and reducing the precision of each parameter. Both methods compress models but likely with a different quality implication. The paper is essentially studying the way of producing the best quality per byte of parameters. When scaling up the models, it uses compound scaling similar to the one used in EfficientNet and standard quantization recipes. Experiments are done on both image classification tasks with convolutional models and language tasks with Transformer model. It is discovered that 2-bit weights provide the best trade-off in most scenarios.

**Strengths:**

The problem studied in this paper --- whether to reduce the number of parameters or to reduce the precision --- is indeed important and challenging for the ML efficiency area. The difficulty is that the lessons learnt from this topic evolve from time to time, especially as the quantization methods evolve. Back in 2022, [1] studied a similar topic on LLMs and concluded that 4-bit is the best as models with below 4-bit precision failed to converge. However in 2023, [2] discovered that 1-bit weight in Transformer will produce a better trade-off. In this paper, 2-bit weights seem the best.

The paper is therefore entering an important and complex area. It provides more evidence on assessing the benefit of low-precision quantization vs. compact models.

[1] The case for 4-bit precision: k-bit Inference Scaling Laws. T. Dettmers et al., ICML 2023.
[2] Binarized Neural Machine Translation. Y. Zhang et al., NeurIPS 2023.

**Weaknesses:**

The concern of this paper is around the quality of experiments. It is detailed as follows:

1. The base model is not verified fully. The paper replaces the SiLU activatioin function in the EfficientNet model used in the experiments with ReLU. The reason is to remove negative outputs with small magnitudes in order to utilize the quantization buckets efficiently. While this is true, it means the results have already been in favor of quantization. In order to be more fair, the paper should at least ablate that replacing the activation function with ReLU does not cause floating-point model quality regression.

2. The model size is less practical. In line 223, it is mentioned that the base model size used in the experiments is 10x smaller than EfficientNet-B0. It is not clear whether such a small model can produce convincing baseline quality.

3. The training setup is less practical. In line 344, it is mentioned that the models are trained for 30 epochs on ImageNet. It is not clear if they are converged.

4. Experiments use mixed quantization algorithms. In line 417, it is mentioned that part of the experiments used LSQ while others use "plain linear quantization". Despite there is no definition of the plain linear quantization algorithm, this mixture incurs additional degree of change, thus requiring ablation.

5. The experiments did not extend to 1-bit due to limit of large model training. This weakens the conclusion of "2-bit is mostly better in trade-off".

6. Presentation issue: In Figure 3, model params/width is not visible. Providing a table with those numbers would be better for assessment.

**Questions:**

Questions are included in the weakness section.

---

### Official Review · Reviewer_tGLV · 2024-11-04

**Soundness:** 2
**Presentation:** 3
**Contribution:** 2
**Rating:** 5
**Confidence:** 3

**Summary:**

This paper contains a comparison of different ways to reduce the complexity of a model: reducing the number of model parameter and reducing the precision of the parameters. It first establishes an objective metric with which the different methods
are compared by, which is the model performance under a memory constraint. The paper arrives at the conclusion that keeping a lot of low precision parameters is preferable to keeping a few high precision ones. Their results are supported with experiments from the domains of language processing and computer vision.

**Strengths:**

- The paper poses a question of high practical relevancy and generates valuable insight for a machine learning practitioner and further research on the topic
- The paper has a very clear research question, well understandable motivation and supporting experiments which cover a sufficiently wide range of use cases

**Weaknesses:**

-  While the paper is aims at answering exactly the question of “more parameters with less precision or vice-versa?”, the paper loses out on practical applicability by not also reporting the run-times of models, which is often a critical bottleneck when considering Ressource-constrained applications such as the edge devices mentioned in the title of the paper
- The LLM experiments on Wikitext are of limited applicability, as the model is extremely small, which might be reflected in the erratic behaviour of the model in the results (as mentioned in the paper). All compressed models perform very poorly compared to the baseline (>40 ppl vs 18 ppl, as reported in appendix A.1.1)
- While evaluation on CIFAR-10 was done very rigorously, the following experiment sections are a bit weaker (no reports for different compression ratios, no reports for width-only compression, which could be included for the following experiments in the same way as was done for CIFAR-10)

**Questions:**

- Please clarify in which cases each value of k is used in the setup (bottom of p. 4)
- Some formatting recommendations (which didn’t negatively impact the review):
1. Write “performance” as non-cursive text or use a shorthand; maybe even rewrite it as a problem of minimising the loss, as is arguably more common in ML setups
2. Use different letters for a and alpha in eq. 2, as it is hard to distinguish while reading, for example a and a^{\ast}
3. Spelling: “In this paper, we attemp[t] to…]” (line 20), “when seeking [an] on-device […] intelligence application[s]” (line 32-33)
- Despite being mentioned in the paper title, I see little connection to edge ai, neither in the introduction nor in the results. Because of this would consider changing the title (f.e. to "Exploring the Trade-Off between Model Complexity and Numerical Precision for Efficient Inference")
- If possible, filling in the experimental gaps mentioned in the weaknesses section would definitely help supporting the claims of the paper

---

### Official Review · Reviewer_sYmn · 2024-11-05

**Soundness:** 1
**Presentation:** 2
**Contribution:** 1
**Rating:** 1
**Confidence:** 5

**Summary:**

This paper studied the behavior of model performance (accuracy, in this work) as a function of number of parameters and precision under a given memory constraint. For example, author first determined the max memory allowed for a given task, and then applied a compression ratio to the full-precision model in a similar way to EfficientNet so that the model can satisfy the memory requirement. Furthermore, author chose to maintain the same memory consumption while varying the numerical precision, e.g. when precision is halved, the number of parameters will be doubled accordingly. Based on the experimental results using ResNet, EfficientNet, and Transformers on vision, NLP, and audio tasks, the author observed a trend that lower numeric precision with higher number of parameters seems to provide better performance.

**Strengths:**

Good amount of QAT experiments on vision, NLP, and audio tasks.

**Weaknesses:**

1. Problem definition unclear.
This study is based on the assumption of constant-memory consumption and then vary the precision and number of parameters accordingly, implying an edge device application with very limited RAM. However, the "memory constraint" was not clearly defined and varied a lot from task to task. For example, In 4.1 CIFAR10 case, the models were chosen to have 270k parameters; in 4.2 ImageNet case, memory budget is set to be 2.6MB which would translate to ~650k parameters in full precision; in 4.3 models is set to have 8.4m parameters; and in 4.4 memory budget is set to 230kB, i.e. only ~57.5k full precision parameters was allowed. Even the similar vision tasks, i.e. 4.1 and 4.2, are using quite different constraints. It would be crucial to clearly state how the memory constraints were chosen so that the following studies could be justified.

2. Generalization
The W32A32 "baseline" data points in this study are mostly far from the original design of the model, e.g. Fig. 4 and 5 had applied an aggressive compression ratio of 10x, 32x, respectively. Naively "compress" a model this way will almost certainly create an useable model, such as in Fig. 4 W32A32 "baseline" shows a top-1 error rate of ~90%. Since the prerequisite of a compression technique is to maintain an acceptable model quality, results from catastrophic failure models should be considered as invalid data points. Plotting data points from failed models and observing trend from mostly invalid points would not lead to any generalizable conclusions.

**Questions:**

see weakness

---

### Meta-Review · Area_Chair_hRfn · 2024-12-20

**Metareview:**

This paper investigates the trade-off between model parameter count and numerical precision under memory constraints, finding that lower precision with a higher parameter count generally yields better performance across various tasks, including image classification, keyword spotting, and language modeling. The paper's strengths lie in its exploration of a highly relevant question for efficient model deployment and its support from a wide array of experiments, spanning different domains and architectures.  However, the reviewers noted that the study suffers from ill-defined and inconsistent memory constraints across tasks, the utilization of aggressively compressed models that are practically unusable, and a lack of consideration for real-world factors like runtime, weakening the generalizability and applicability of its conclusions. Therefore, while the work touches upon an important topic, it is not suitable for publication at this time.

**Additional Comments On Reviewer Discussion:**

The authors did not provide a rebuttal and there was no discussion.

---

### Decision · Program_Chairs · 2025-01-22

Reject